# Probability density and information entropy of machine learning derived intracranial pressure predictions

**Anmar Abdul-Rahman**[1]*, **William Morgan**[2,3], **Aleksandar Vukmirovic**[2,3], **Dao-Yi Yu**[2,3]

**1** Department of Ophthalmology, Counties Manukau District Health Board, Auckland, New Zealand, **2** Centre for Ophthalmology and Visual Science, The University of Western Australia, Perth, Australia, **3** Lions Eye Institute, University of Western Australia, Perth, Australia

* anmar_rahman@hotmail.com

**Data Availability Statement:** All relevant data are within the manuscript and its Supporting information files.

## Abstract

Even with the powerful statistical parameters derived from the Extreme Gradient Boost (XGB) algorithm, it would be advantageous to define the predicted accuracy to the level of a specific case, particularly when the model output is used to guide clinical decision-making. The probability density function (PDF) of the derived intracranial pressure predictions enables the computation of a definite integral around a point estimate, representing the event's probability within a range of values. Seven hold-out test cases used for the external validation of an XGB model underwent retinal vascular pulse and intracranial pressure measurement using modified photoplethysmography and lumbar puncture, respectively. The definite integral ±1 cm water from the median ($DI_{ICP}$) demonstrated a negative and highly significant correlation (-0.5213±0.17, p< 0.004) with the absolute difference between the measured and predicted median intracranial pressure ($Diff_{ICPmd}$). The concordance between the arterial and venous probability density functions was estimated using the two-sample Kolmogorov-Smirnov statistic, extending the distribution agreement across all data points. This parameter showed a statistically significant and positive correlation (0.4942±0.18, p< 0.001) with $Diff_{ICPmd}$. Two cautionary subset cases (Case 8 and Case 9), where disagreement was observed between measured and predicted intracranial pressure, were compared to the seven hold-out test cases. Arterial predictions from both cautionary subset cases converged on a uniform distribution in contrast to all other cases where distributions converged on either log-normal or closely related skewed distributions (gamma, logistic, beta). The mean±standard error of the arterial $DI_{ICP}$ from cases 8 and 9 (3.83±0.56%) was lower compared to that of the hold-out test cases (14.14±1.07%) the between group difference was statistically significant (p<0.03). Although the sample size in this analysis was limited, these results support a dual and complementary analysis approach from independently derived retinal arterial and venous non-invasive intracranial pressure predictions. Results suggest that plotting the PDF and calculating the lower order moments, arterial $DI_{ICP}$, and the two sample Kolmogorov-Smirnov statistic may provide individualized predictive accuracy parameters.

**Funding:** The author(s) received no specific funding for this work.

**Competing interests:** I have read the journal's policy and the authors of this manuscript have the following competing interests: We would like to declare that the authors Anmar Abdul-Rahman, William Morgan, and Dao-Yi Yu are the inventors of the Modified Photoplethysmography method. Furthermore, we have no financial interest in the results of this study. This does not alter our adherence to PLOS ONE policies on sharing data and materials.

## Introduction

Although machine learning models can be valuable in providing recommendations, the lack of a human factor in the decision-making process and the algorithm's "black box" nature risk creating a false authority impervious to appeal or rational argument. Uncertainty can be mitigated by statistical goodness of fit parameters, qualitative and quantitative extensions can assist decision-making. Conventionally, mean squared error (MSE) and its rooted form root mean squared error (RMSE), as well as the mean absolute error (MAE) and its percentage variant (MAPE), are commonly used as statistical goodness of fit parameters for regression models. Whereas informative, these measures share a common disadvantage: a single-point estimate does not represent the regression performance concerning the distribution of the ground truth element [1]. Another parameter, the coefficient of determination ($R^2$), despite being invariant to linear transformations of the independent variables' distribution, is dependent on the slope of the regression, and this may mask a wider confidence interval around the regression surface in the presence of a steep regression slope [2]. In this framework, it is crucial to establish an individualized parameter of reliability, particularly when depending on the model's output for clinical decision-making. In previous work [3], the measures of central tendency [mean, median, peak density (mode)] were used as point estimates of Extreme Gradient Boost (XGB) derived intracranial pressure (ICP). These predictions were generated from all vascular pulsation points in the image field. High agreement was observed between the median predicted with measured intracranial pressure, where the arterial and venous Bland-Altman bias±standard error was 0.1386±1.6545 and 0.0343±1.8013 cm water, respectively. In addition to the benefits of visualizing the probability density distribution, defining the properties of the distribution would provide further evidence of predictive accuracy.

There are three possibilities for the distributions of XGB-derived probability density functions. If the probability density function (PDF) converged upon a symmetrical unimodal distribution (Gaussian, beta), then the distribution weights, by definition, should coincide [4, 5]. Other symmetrical unimodal distributions (triangular and Cauchy distributions) are not likely, as triangular distributions are not consistent with biological characteristics, and the latter has infinite integral without finite moments of order greater than or equal to one; only fractional absolute moments exist [6, 7]. Although a symmetrical unimodal distribution would be convenient, as the central tendencies would coincide, this instance is unlikely. Skewed distributions (log-normal, beta, gamma, and exponential) are common when mean values are low, variances large, and variables cannot be negative [8]. These represent the most likely candidates for the distribution of the XGB-derived intracranial pressure, where the distribution of central tendencies demonstrated a range of agreements with measured intracranial pressure [3]. Finally, a uniform PDF can exist if predicted intracranial pressure is no better than random.

Although visualization offers a qualitative assessment of the PDF; a quantitative approach would include plotting the PDF and calculating the moments of the distribution. A PDF defines the relationship between a continuous random variable (x) and its probability distribution f(x). This function has four properties [9]:

1. The function must be greater than zero: f(x) $>$0

2. The area under the curve must be equal to one: $\int_{-\infty}^{\infty} f(x)\,dx = 1$

3. The function f(x) is piecewise continuous.

4. The definite integral between two values represents the probability of occurrence of the variable between two points [a,b]: $P(a < X \leq b) = \int_{a}^{b} f(x)dx$

Machine learning has a wide range of possible applications where predicting the probability density for a variable rather than a point estimate would be more informative. This is either because variance is input-dependent or small probabilities, although maybe analytically trivial, represent significant real-world outcomes [10, 11]. This is the case for intracranial pressure estimation, where a small margin of error would result in a significant difference in a clinical outcome. There is a wide variability in normal intracranial pressure values largely due to differences in age, gender, and body mass index. Although normal values of 7–15mmHg have been reported [12]. In a prospective study of lumbar puncture measured ICP in 339 normal subjects, Bø et al. reported a reference range of approximately 3–22mmHg [13]. Their findings suggest that physiological ICP may vary by up to a 7-fold range. Additionally studies demonstrate variation of continuously measured ICP within individuals, likely due to the multitude of interactions between physiologic parameters involved in intracranial pressure homeostasis, including postural, cardiovascular, neurological and respiratory parameters [14]. In a systematic review and meta-analysis of ICP monitoring systems Zaccharetti et al. found that the average error between simultaneous ICP measurements from different pressure sensors was approximately 1.5 mmHg, but the variability was large, with up to 11.4 mmHg difference in 95% of readings [15]. These factors render determining the precision and accuracy of ICP measurements challenging even in absence of underlying pathology. While evidence suggests artificial intelligence (AI) algorithms could aid clinical decision-making, current research emphasizes the need for systems enabling model evaluation, bias detection, and generalizability. Wang et al. compared machine learning models to traditional scoring tools for predicting mortality risk in traumatic brain injury patients. Among 47 studies with 156 models, machine learning models showed relatively high accuracy for both in-hospital and out-of-hospital mortality. Notably, traditional tools achieved comparable accuracy. The authors highlight the need for standardized reporting and validation of machine learning models to ensure clinical applicability and generalizability [16]. Similarly, van Hal et al. assessed bias risk and clinical readiness of studies using AI to predict intracranial hypertension in traumatic brain injury patients. They found most studies had high bias risk and low readiness for clinical integration. Despite promising potential, the authors concluded further improvement and validation are necessary before implementing these models in clinical practice [17]. These findings underline the practical importance of bias detection, as AI algorithms may perform well under specific conditions not replicable in clinical settings. Using properties of the probability density function the approximation of the estimate to the ground truth may be confirmed by the capacity to validate ICP readings, irrespective of the measurement methodology.

The other distinct advantage of this approach is that the probability density function is normalized, enabling a direct comparison between distributions from different samples [18]. Furthermore, it allows the calculation of the probability of the point estimate within a range of values by computing the definite integral, which is defined as the area under the curve between two arbitrary points [a,b].

Quantitative tests, which include the Shapiro-Wilk, Kolmogorov-Smirnov, Lilliefors, Anderson-Darling, and Cramer-von Mises statistics, among others, can either evaluate the convergence of a variable on a normal distribution in the case of a single sample test or measure the concordance between two distributions in the case of a two-sample test [19, 20]. Among the most commonly employed tests, the Kolmogorov-Smirnov statistic is weighted to the center of the distribution, and the Anderson-Darling statistic is more sensitive to the tails [21, 22]. A distinct advantage of the quantitative approach is that it can estimate the concordance of intracranial pressure probability density distributions derived from the arterial and venous models. The uncertainty (randomness) in a distribution can likewise be evaluated by computing the Shannon entropy (sEnt), also known as information entropy, which increases

with increasing uncertainty in the probability density distribution [23]. A single test statistic can be used to compare the model accuracy across a range of predictions rather than over a single-point estimate. In contrast to a global goodness of fit score, evaluation of the predicted intracranial pressure distribution properties should provide a qualitative and quantitative assessment of the XGB model performance at the level of an individual case. Additionally, this approach could define imaging parameters associated with favorable predictive outcomes.

This work compares the pulsation amplitude decomposition in the frequency domain and properties of the probability density distributions of the hold-out test cases from a previously published work [3] with two cautionary subset cases, 8 and 9. The latter two cases are examples for which the XGB model provided contradictory intracranial pressure predictions.

## Materials and methods

Participants were referred to the Lions Eye Institute over six years (2015–2021) from the Neurology and Neurosurgery Departments and were to undergo lumbar puncture for suspicion of idiopathic intracranial hypertension. Written consent was obtained from each participant. Recruitment for the study occurred between 03/03/2016 -28/11/2021. Several studies have revealed that over 90% of patients with idiopathic intracranial hypertension are female, making gender a well-recognized risk factor in this condition [24], hence the gender bias in this study. Optic nerve color photography combined with modified photoplethysmography, a form of ophthalmodynamometry, was performed on all participants. Lumbar puncture was performed on patients within three days after the modified photoplethysmography. This device consists of a force transducer surrounding a central contact lens, which allows imaging of the optic disc under a dynamic range of induced intraocular pressure ($IOP_i$). Study approval was granted by the University of Western Australia Human Ethics Committee (Approval #: 2015–11-756-A-2.), adhering to the tenets of the Declaration of Helsinki. Participants were to have no prior history of retinal or optic nerve pathology and were required to have transparent ocular media. The Extreme Gradient Boost Model (XGB) was derived from a training/test set of 21 subjects. A total of 129,600 data points were sampled from the images, 56,932 arterial and 72,668 venous data points. A 80/20 training/test data split was implemented. A further 7 subjects were used as hold-out test cases in evaluating this model. Further details of the model are published in previous work [3]. In the current study a total of 9 subjects were included in the analysis in this analysis, 7 subjects were in the hold-out test group from the original dataset (cases 1 to 7), and 2 subjects (cases 8 and 9) were in the cautionary subset. The latter subset demonstrated a wide difference between measured and predicted intracranial pressure and conflicting predictions from the arterial and venous models were found in ongoing evaluation of the model.

The distributions of the harmonic regression wave amplitude ($HRW_a$) and most of the Fourier coefficients were non-normal. Therefore the median was used to measure central tendency, and dispersion was estimated using the interquartile range (IQR); additionally, the range was computed. Hypothesis tests were conducted using the Wilcoxon test with Bonferroni correction. The probability density function for each arterial and venous model was plotted. The definite integral ($DI_{ICP}$) was calculated for all cases by estimating the definite integral ±1 cm water from the median of the predicted intracranial pressure. Computation of $DI_{ICP}$ was confirmed by calculating the probability between the minimum and maximum bounds of the PDF, which returned a value of 1.00 with absolute error < 0.00012 for all cases. The Kolmogorov-Smirnov statistic (KS) and Anderson-Darling statistic (ADS) were used as a quantitative measure of the deviation of the PDF from a normal distribution, the former being more sensitive to the body and the latter to the tails of the distribution [19]. The two-sample

Kolmogorov-Smirnov statistic (tsKS) was calculated as a measure of the magnitude of the distributional difference between arterial and venous PDFs. The larger the test statistic, the more significant the difference between the two distributions. The maximum distance between the empirical cumulative distribution functions was used to visualize the result [25]. The difference between median ICPs ($\text{Diff}_{\text{ICPmd}}$) was calculated as a measure of agreement between predicted and measured intracranial pressure. Multivariate analysis of variance (MANOVA) was used in the assessment of the statistical significance of the difference in the means between $\text{Diff}_{\text{ICPmd}}$ and imaging variables, including test laterality, the levels of $\text{IOP}_i$, the number of data points evaluated in each image, and the model type. Together with the central tendency distribution characteristics including standard deviation (sd), kurtosis (indicating tail weight and not peakedness) [26], and skew (corresponding to the first to the fourth moments). Additionally, $\text{DI}_{\text{ICP}}$, ADS, KS, distribution type as determined using Cullen-Frey graphs, and case subtype (cautionary subset/hold-out test). Shannon entropy, also known as information entropy, is a concept from information theory that quantifies the uncertainty, in this case, associated with a probability density distribution. Shannon entropy of the PDF was computed using the entropy library from R statistical package [27]. It is defined mathematically as:

$$H(x) = -\sum_x P(x)\log_2(P(x))$$

Where:

H(x) is the entropy of the random variable x.

P(x) represents the probability of a specific outcome x in the distribution.

The summation ($\Sigma$) is taken over all possible outcomes in the distribution.

Polychoric and Pearson correlations (for discrete and continuous variables, respectively) were used to examine the relationship between $\text{Diff}_{\text{ICPmd}}$ on one side and imaging and distribution parameters on the other for the venous and arterial models separately. A p-value $<0.05$ was considered statistically significant for all tests. Fig 1 is the workflow schematic.

## Results

Modified photoplethysmography was performed bilaterally in five cases (10 eyes) and unilaterally in four cases (4 eyes). Both cases in the cautionary subset had unilateral imaging performed. Whereas the mean of $\text{Diff}_{\text{ICPmd}}$ in the bilateral tested group was 3.07±0.56, this increased significantly in the unilateral tested group to 14.54±4.37, p<0.01. A total of 19,905 data points were sampled from the images of the study group: 7,617 arterial and 12,288 venous data points. The median measured intracranial pressure was 22 cm water, and the IQR was 10 cm water. The distribution was skewed to the left (skew = -0.278, kurtosis = 1.894). Therefore, the weight of the distribution of pulsation data features was at an ICP $\geq$22cm water.

The median, IQR of the $\text{HRW}_a$ in the hold-out test group was higher in the retinal veins (6.123, 0.38) compared to the retinal arteries (4.929, 0.44), p<0.0001. Further details of the Fourier coefficients subsetted by vessel type and study group are listed in Table 1. A violin plot comparing the statistical properties of the $\text{HRW}_a$ is demonstrated in Fig 2.

Table 2 and the ridgeline plot (Fig 3) provide a summary of the characteristics and probability density distributions for the predicted intracranial pressure for hold-out test and the cautionary subset (cases 8 and 9) cases. It can be observed that $\text{Diff}_{\text{ICPmd}}$ for the latter two cases was at the higher limit for both XGB models compared to the hold-out test cases. The arterial $\text{Diff}_{\text{ICPmd}}$ was (22.22, 37.09) and venous (15.01, 23.20) for cases 8 and 9, respectively. Of the parameters of the probability density distribution, the arterial $\text{DI}_{\text{ICP}}$ in case 8 was 3.27% and case 9 was 4.39%, both lower (mean±standard error, 3.83±0.56%, sd = 0.792) than the $\text{DI}_{\text{ICP}}$

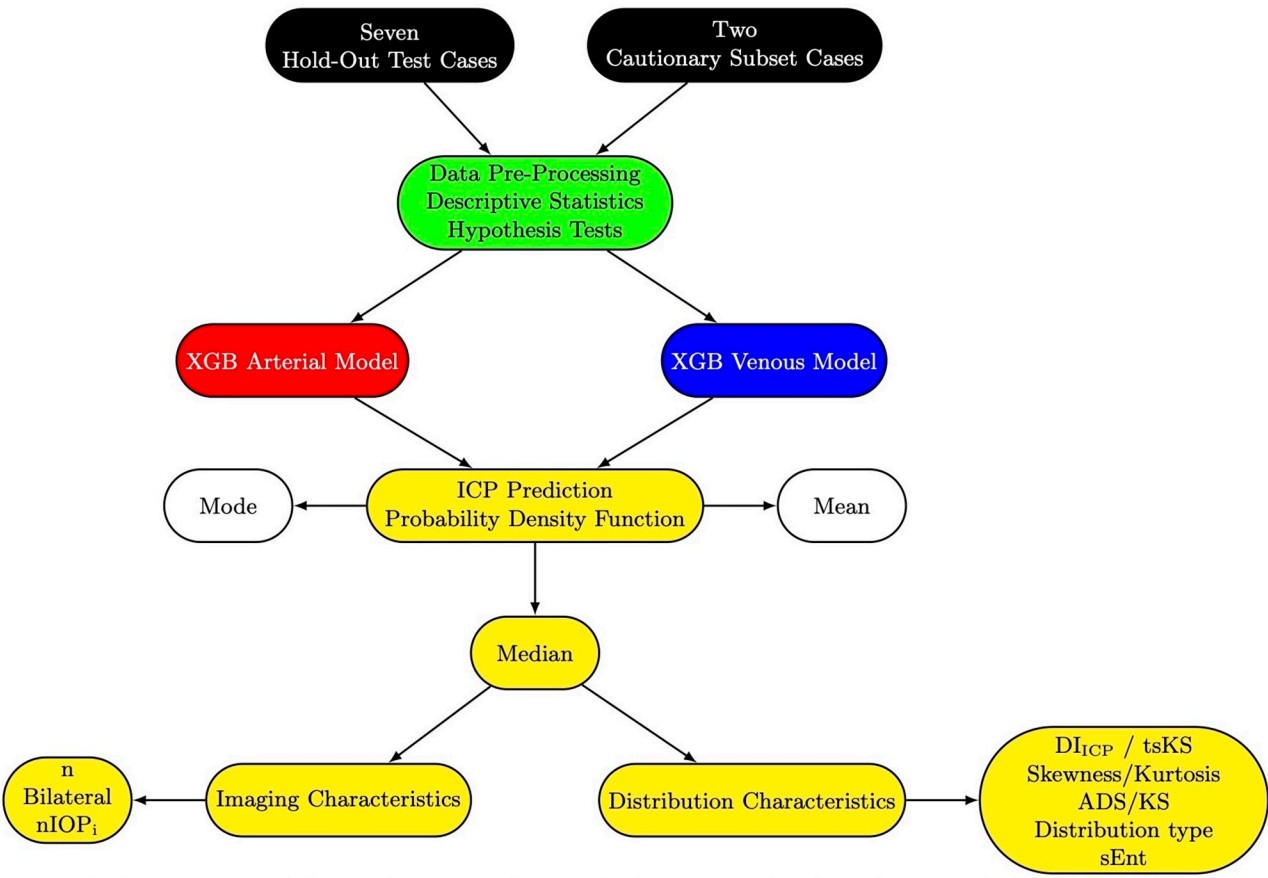

**Fig 1. Data workflow schematic for probability density function analysis of Extreme Gradient Boost derived intracranial pressure predictions.**
Two cautionary subset cases (8 and 9) were subsetted from seven hold-out test cases and demonstrated a wide and conflicting difference between measured and predicted intracranial pressure from the arterial and venous XGB models. All cases underwent data pre-processing, descriptive statistics, and hypothesis tests were computed from the retinal vascular pulse parameters. Intracranial pressure predictions were derived from the retinal arterial and venous parameters independently. Probability density functions were generated from intracranial pressure predictions from both XGB models, where the median was considered the most favorable compared to the mean and the mode. This was likely because the median represents the geometric mean of a log-normal distribution and is supported by findings from previous work [3]. Correlations were computed between the absolute difference between the median predicted and measured intracranial pressure ($\text{Diff}_{\text{ICPmd}}$) and imaging characteristics: (n = number of vascular data points analyzed, Bilateral = both eyes tested, $\text{nIOP}_i$ = number of induced intraocular pressure levels applied during imaging) and distribution characteristics: ($\text{DI}_{\text{ICP}}$ = definite integral ± 1cm water of the median, tsKS = two-sample Kolmogorov-Smirnov statistic, ADS = Anderson-Darling statistic, KS = Kolmogorov-Smirnov statistic), sEnt = Shannon entropy.

computed from all hold-out test cases (14.14±1.07%, p<0.03, sd = 2.5). The venous $\text{DI}_{\text{ICP}}$ in case 8 was 24.46%, and that of case 9 was 8.22%, both >2 standard deviations from the mean $\text{DI}_{\text{ICP}}$ of the hold-out test cases, whereas the median (IQR) for sd, kurtosis, and skew in the hold-test cases was 7.852 (2.250), 4.845 (1.835), and 1.0850 (0.81), indicating a leptokurtic distribution (kurtosis>3), the values for the cautionary cases were 11.843 (4.861), 2.425 (1.1025), and -0.0450 (0.445) indicating a platykurtic distribution (kurtosis<3). When the distributions were superimposed as shown in Fig 4, it can be observed that cases with a close overlap of arterial and venous probability density distributions intuitively have close predictive outcomes (Table 2).

Ridgeline plot of the probability density distributions observed in Fig 3 is derived from the arterial and venous XGB models. It highlights the convergence on skewed distributions for most predictions. The level of concordance between arterial and venous distributions can be

**Table 1. Descriptive statistics of the harmonic regression wave amplitude, the cosine and sine coefficients of the first and second harmonics of the hold-out test and cautionary subset cases.**

|  | Site | Median | IQR | Min | Max | Range | Skew | Kurtosis |
|---|---|---|---|---|---|---|---|---|
| **Hold-Out** |  |  |  |  |  |  |  |  |
| $HRW_a$ | Artery | 6.186 | 4.929 | 0.44 | 43.501 | 43.061 | 1.747 | 4.765 |
| $a_{n1}$ | Artery | -0.526 | 2.988 | -9.462 | 10.746 | 20.208 | -0.448 | 0.963 |
| $b_{n1}$ | Artery | -2.049 | 1.828 | -15.195 | 6.387 | 21.582 | -1 | 4.077 |
| $a_{n2}$ | Artery | 0.318 | 1.026 | -3.124 | 5.96 | 9.083 | 0.899 | 2.352 |
| $b_{n2}$ | Artery | 0.05 | 0.878 | -4.221 | 9.6 | 13.821 | 0.783 | 6.561 |
| $HRW_a$ | Vein | 7.028 | 6.123 | 0.38 | 64.241 | 63.861 | 2.865 | 11.725 |
| $a_{n1}$ | Vein | 1.664 | 2.391 | -22.936 | 21.63 | 44.566 | 0.995 | 7.928 |
| $b_{n1}$ | Vein | -2.24 | 2.172 | -25.153 | 11.503 | 36.657 | -2.769 | 14.537 |
| $a_{n2}$ | Vein | -0.228 | 1.055 | -7.551 | 8.557 | 16.109 | -0.273 | 4.622 |
| $b_{n2}$ | Vein | 0.203 | 0.815 | -13.343 | 10.293 | 23.635 | 0.389 | 22.033 |
| **Subset** |  |  |  |  |  |  |  |  |
| $HRW_a$ | Artery | 4.099 | 2.256 | 1.391 | 14.847 | 13.456 | 1.566 | 3.716 |
| $a_{n1}$ | Artery | -0.954 | 0.748 | -5.429 | 1.522 | 6.951 | -0.552 | 3.356 |
| $b_{n1}$ | Artery | -1.304 | 0.847 | -4.257 | 1.387 | 5.644 | -0.892 | 1.865 |
| $a_{n2}$ | Artery | 0.338 | 0.624 | -1.081 | 2.133 | 3.215 | 0.235 | 0.285 |
| $b_{n2}$ | Artery | -0.046 | 0.817 | -2.383 | 2.065 | 4.448 | -0.181 | 0.424 |
| $HRW_a$ | Vein | 6.127 | 4.425 | 1.306 | 32.956 | 31.65 | 2.093 | 4.68 |
| $a_{n1}$ | Vein | 1.197 | 3.466 | -5.257 | 12.665 | 17.922 | 0.984 | 1.476 |
| $b_{n1}$ | Vein | -2.102 | 1.777 | -11.201 | 2.942 | 14.144 | -1.889 | 4.507 |
| $a_{n2}$ | Vein | -0.039 | 0.754 | -2.578 | 1.929 | 4.506 | -0.621 | 1.102 |
| $b_{n2}$ | Vein | 0.367[b] | 0.952 | -2.368 | 4.242 | 6.61 | 0.559 | 1.404 |

$HRW_a$ = harmonic regression wave amplitude, $a_{1,2}$ = Fourier cosine coefficient of the first and second harmonic, $b_{n1,2}$ = Fourier sine coefficient of the first and second harmonic, IQR = interquartile range, Highlighted cells[b] = between group median differences failed to achieve statistical significance at a level $p<0.05$.

quantitatively estimated in Fig 4. The distributions are superimposed, and the tsKS was computed. The lower the value of tsKS, the higher the concordance between the distributions. For all cases, the value of tsKS was statistically significant. The exemplar case (Case 4) demonstrated the lowest tsKS statistic (0.080897, $p<0.003$), although Case 7 had the highest tsKS statistic (0.48101). This was offset by lower $Diff_{ICPmd}$ (arterial 0.41, venous 7.56). For this case, both venous and arterial distributions were multimodal and converged on the broader family of beta distributions. Cases 8 and 9 showed tsKS statistics values towards the higher end (0.47723 and 0.3755, respectively), confirming the lack of concordance between arterial and venous probability density distributions (Fig 5).

Multivariate analysis of variance (MANOVA) was used to evaluate the statistical significance of the relationship between $Diff_{ICPmd}$ and both imaging and model output distribution characteristics. The results are summarized in Table 3. It can be observed that of the imaging characteristics, both laterality and the number of induced intraocular pressure levels ($nIOP_i$). Interestingly the number of data points analyzed failed to achieve statistical significance. Of the single distribution parameters, those that defined the shape of the distribution (sd, skew, kurtosis, $DI_{ICP}$) demonstrated significant associations, as did the concordance measure for between distribution (tsKS) agreement. In contrast, parameters that indicated convergence upon a normal distribution (ADS, KS) were insignificant. These results suggest that bilateral modified photoplethysmography undertaken under multiple levels of induced intraocular pressure is more likely to generate probability density distributions with favorable characteristics.

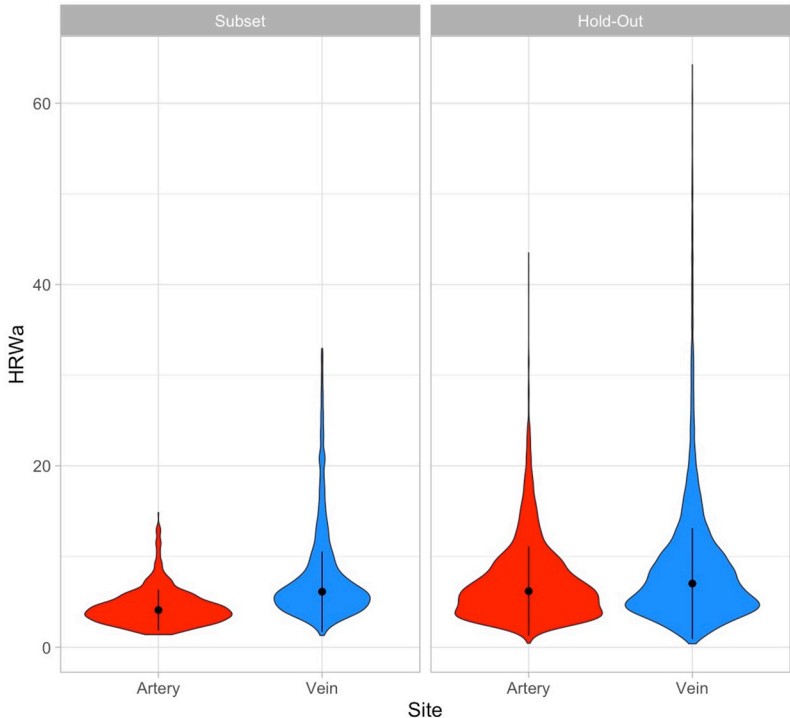

**Fig 2. Violin plot.** Comparing the distribution of the harmonic regression wave amplitude in the hold-out and cautionary subsets.

The associations can be further clarified by evaluating the strength and direction of the correlations by computing the correlation statistic. This can be visualized in the correlation matrix in Fig 6. Five conclusions can be drawn from the matrix:

1. The importance of bilateral imaging and recording the pulsation characteristics under multiple levels of $IOP_i$ was supported by a highly significant and negative correlation statistic (-0.59, -0.49).

2. Except for (sd), which was significantly and positively correlated, distribution shape parameters (skew, kurtosis, $DI_{ICP}$) were all significantly and negatively correlated with $Diff_{ICPmd}$. The negative correlation indicates that a right-skewed narrower distribution with fewer outliers favors a more accurate XGB prediction.

3. The higher the concordance between arterial and venous XGB distributions, the more accurate the prediction. The positive correlation is indicative that both parameters need to be smaller for more favorable predictions.

4. Laterality was highly and negatively correlated with tsKS (-0.98) and moderately and positively correlated with $DI_{ICP}$ (0.33). This result suggests that bilateral imaging was significant for analyzing the concordance measure (tsKS). In contrast, with $DI_{ICP}$, a weaker strength of association indicated that this measure was less affected by test laterality.

5. A comparison of the four methods estimating randomness in the PDF, correlations with $Diff_{ICPmd}$ decreased in the following order: lower order distribution moments (sd, skew, kurtosis), $DI_{ICP}$, tsKS, and sEnt.

**Table 2. Imaging characteristics and probability density distribution parameters of hold-out test cases 1–7 and cautionary cases 8 and 9 are compared to measured and Extreme Gradient Boost predicted intracranial pressure (cm water).**

| Case | ICP | n | nIOPi | Lat | Mean | Median | Mode | sd | Skew | Kurtosis | DImd | ADS | KS | D | Diffmd | sEnt |
|---|---|---|---|---|---|---|---|---|---|---|---|---|---|---|---|---|
| **Arterial** | | | | | | | | | | | | | | | | |
| 1 | 22 | 2348 | 14 | 2 | 26.59 | 25.18 | 24.98 | 8.251264 | 1.05 | 4.56 | 13.95 | 15.808 | 0.057 | G | 3.18 | 5.587466 |
| 2 | 32 | 248 | 9 | 1 | 24.13 | 24.05 | 24.45 | 4.997260 | 0.7 | 6.02 | 16.69 | 1.144 | 0.06 | L | 7.95 | 5.381761 |
| 3 | 20 | 701 | 13 | 2 | 24.52 | 21.63 | 20.07 | 8.060139 | 1.3 | 4.12 | 15.19 | 14.455 | 0.124 | Bt | 1.63 | 5.555466 |
| 4 | 32 | 1026 | 15 | 2 | 31.86 | 30.66 | 28.7 | 8.229153 | 0.32 | 2.81 | 10.08 | 1.993 | 0.034 | Bt | 1.34 | 5.657664 |
| 5 | 17 | 1396 | 12 | 2 | 23.26 | 22.8 | 21.36 | 5.797218 | 0.52 | 5.49 | 14.72 | 6.047 | 0.064 | L | 5.80 | 5.171197 |
| 6 | 27 | 1447 | 15 | 2 | 28.25 | 25.12 | 24.05 | 8.797857 | 1.43 | 4.76 | 17.63 | 36.552 | 0.142 | L | 1.88 | 5.511104 |
| 7 | 26 | 446 | 7 | 1 | 28.65 | 25.59 | 23.38 | 8.791855 | 0.76 | 2.94 | 10.7 | 5.401 | 0.102 | Bt | 0.41 | 5.688154 |
| 8 | 11 | 358 | 9 | 1 | 33.05 | 33.22 | 44.1 | 13.307430 | -0.11 | 1.83 | 3.27 | 8.424 | 0.141 | U | 22.22 | 5.795639 |
| 9 | 8 | 653 | 9 | 1 | 41.82 | 45.09 | 39.09 | 14.859540 | 0.02 | 2.18 | 4.39 | 6.426 | 0.085 | U | 37.09 | 5.848207 |
| **Venous** | | | | | | | | | | | | | | | | |
| 1 | 22 | 2020 | 14 | 2 | 27.77 | 26.34 | 24.47 | 8.522667 | 1.83 | 8.04 | 12.39 | 11.76 | 0.056 | G | 4.34 | 5.353739 |
| 2 | 32 | 787 | 9 | 1 | 30.9 | 29.13 | 27.77 | 9.451438 | 1.59 | 7.22 | 12.53 | 5.869 | 0.069 | LN | 2.87 | 5.46967 |
| 3 | 20 | 837 | 13 | 2 | 24.18 | 23.21 | 22.98 | 6.102340 | 1.5 | 6.73 | 17.68 | 3.651 | 0.056 | LN | 3.21 | 5.392312 |
| 4 | 32 | 936 | 15 | 2 | 31.29 | 30.55 | 30.2 | 7.643098 | 0.55 | 4.19 | 13.47 | 4.784 | 0.071 | LN | 1.45 | 5.548359 |
| 5 | 17 | 1936 | 12 | 2 | 22.78 | 23 | 23.33 | 5.714698 | 0.29 | 4.93 | 17.84 | 38.071 | 0.098 | L | 6.00 | 5.353076 |
| 6 | 27 | 3617 | 15 | 2 | 26.26 | 25.09 | 24.01 | 6.510826 | 1.3 | 5.83 | 15.79 | 13.964 | 0.037 | L | 1.91 | 5.48403 |
| 7 | 26 | 742 | 7 | 1 | 20.7 | 18.44 | 15.91 | 6.921495 | 1.12 | 3.98 | 12.97 | 11.765 | 0.099 | Bt | 7.56 | 5.465993 |
| 8 | 11 | 939 | 9 | 1 | 25.77 | 26.01 | 26.17 | 4.203027 | -0.58 | 4.77 | 24.46 | 23.953 | 0.116 | L | 15.01 | 5.398153 |
| 9 | 8 | 474 | 9 | 1 | 33.67 | 31.2 | 27.41 | 10.37868 | 0.81 | 2.67 | 8.22 | 12.124 | 0.147 | Bt | 23.20 | 5.592257 |

Measures of central tendency (mean, median, mode) of predicted intracranial pressure (ICP) are compared with lumbar puncture measured ICP. From previous work the median venous predicted ICP provided the best agreement with measured ICP [3]. The cautionary cases (8 and 9) demonstrate lower $DI_{ICP}$ and higher DiffICPmd. Although there is a tendency towards skewed distributions (D), the arterial XGB output from both cautionary cases converged on uniform distributions. n = number of data points tested, $nIOP_i$ = total number of induced intraocular pressure measurements performed, Lat = (2) bilateral, (1) unilateral modified photoplethysmography undertaken, $DI_{ICP}$ = definite integral within ± 1 cm water of the median of the probability density function, ADS = Anderson-Darling statistic, KS = Kolmogorov-Smirnov statistic, $Diff_{ICPmd}$ = absolute difference between measured and median predicted intracranial pressure, sd = standard deviation, sEnt = Shannon entropy, G = Gamma, L = Logistic, LN = Log-normal, U = Uniform, Bt = Beta.

Shannon entropy varied over a narrow range for both models. The mean (range) for the arterial model was 5.577 (5.171–5.848) compared to the venous model 5.451 (5.353–5.592). The Pearson correlation between sEnt and $Diff_{ICPmd}$ was 0.4832±0.1788 (p < 0.002). Although Pearson correlation did not achieve statistical significance when subsetted by vascular model, significance of sEnt as a discriminating parameter was suggested by distinguishing convergence of the PDF to a uniform distribution, where sEnt is maximised when the models were aggregated (Table 3).

Pearson correlations evaluating the relationship between $Diff_{ICPmd}$ and $DI_{ICP}$ showed a negative correlation between these two parameters (-0.5213, p< 0.004). The significance persisted for the arterial model only (0.75±0.20, p<0.007). In contrast, the venous model did not achieve statistical significance (0.79±0.17, p<0.58) for this correlation as demonstrated in Fig 7. Similarly, neither the mean KS nor ADS achieved statistical significance in either model (p<0.09). Other parameters of the XGB derived probability density, in particular the distribution type, sd, kurtosis, and skew, were significant in both the MANOVA model and in their polychoric correlations with $Diff_{ICPmd}$ (p<0.003). These differences did not persist when tested by the vascular model subtype. This is possibly due to the limited size dataset. The two-

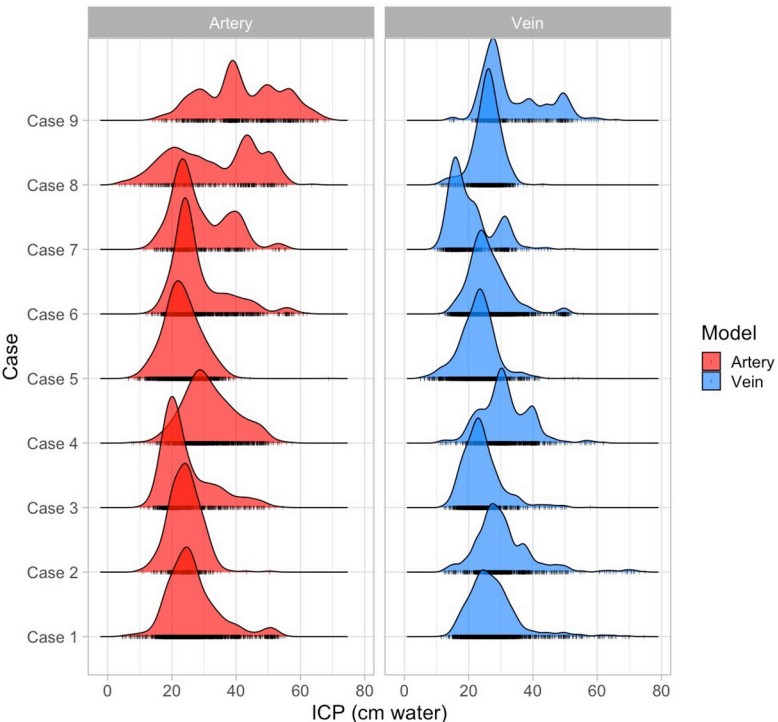

**Fig 3. Ridgeline plot for the probability density function of intracranial pressure predictions derived from the Extreme Gradient Boost algorithm of the arterial and venous pulsation data.** In contrast to the dominant right-skewed distribution in most cases, in cases 8 and 9, the distribution of the arterial predictions converges on a uniform distribution.

sample Kolmogorov-Smirnov statistic showed a statistically significant and positive correlation ($0.4942 \pm 0.18$, $p < 0.001$) with $\text{Diff}_{\text{ICPmd}}$, however, the statistical significance was not sustained when tested for the arterial and venous models independently.

While in the venous model, one-third of the predictions (33.33%) converged on a log-normal distribution, the dominant distributions in the arterial model were logistic and beta distributions (33.33% each) except for both arterial models of cases 8 and 9, which converged on a uniform distribution this finding indicated that the point estimate from the arterial prediction was no better than random in these two cases. Furthermore, this distribution contributed to both cases' low arterial $\text{DI}_{\text{ICP}}$ values. Notably, the venous and arterial distributions in case 8 demonstrated negative skews, a departure from the skews of all other cases. In case 9, the venous distribution converged on a heavy-tailed beta distribution, contributing to the low venous $\text{DI}_{\text{ICP}}$ value in this case.

## Discussion

Despite a limited-sized dataset precluding regression analysis, the results indicated that agreement between the arterial and venous models could be evaluated qualitatively by assessment of the overlap between the probability density distributions or quantitatively by computing the tsKS statistic. Additionally, measures of distribution moments consistent with a narrow-shaped leptokurtic distribution (narrow sd, positive skewness, high kurtosis, and high $\text{DI}_{\text{ICP}}$) supported favorable predictions. Quantifying the uncertainty in the PDF by estimating the Shannon entropy, could provide a further indication of convergence to a uniform distribution

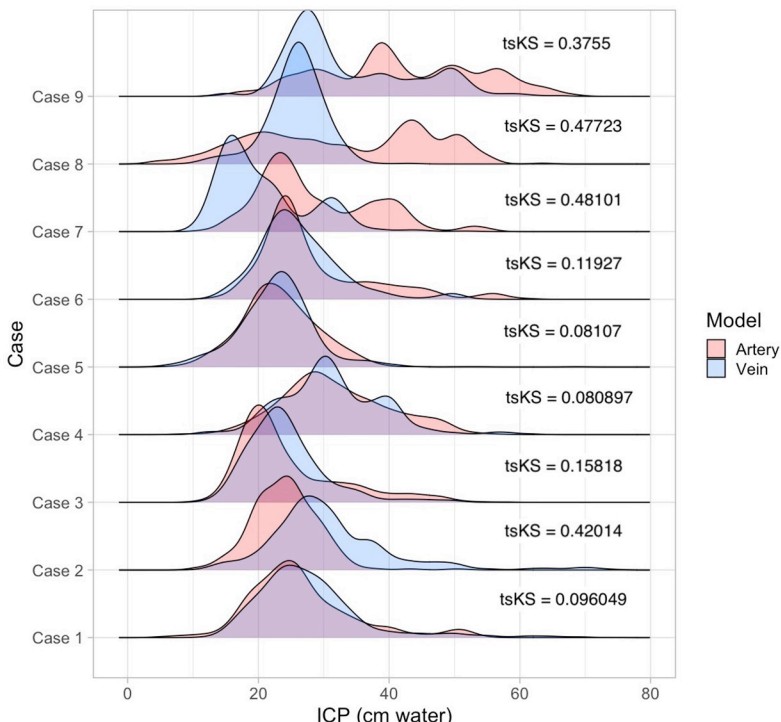

**Fig 4. Overlapping ridgeline plots of intracranial pressure predictions derived from the Extreme Gradient Boost algorithm of the arterial and venous pulsation data.** The two-sample Kolmogorov-Smirnov statistic (tsKS) provides a quantitative comparison between two distributions across the whole range rather than just a point estimate. Within a single case, the closer the approximation of the distributions from the arterial and venous models, the lower the tsKS value. Case 4 demonstrates the lowest tsKS statistic (0.080897, p<0.003), and case 7 is the highest (0.48101).

where entropy is maximised, hence indicating degradation of ICP predictions. However, lower correlation of sEnt compared to other measures can be observed. This may be due to its sensitivity to independence of probability events. If events are not independent the joint entropy can be less than the sum of the individual entropies thereby reducing the value for heavy tailed non-uniform probability density distributions.

Intuitively imaging both eyes and applying multiple $IOP_i$ improved the predictive outcome. A quantitative comparison of the two distributions by the tsKS test is highly advantageous as its performance is independent of the distribution type. It has been applied as a measure of model classification in several other studies [28–30]. Its limitations include restriction to continuous distributions and higher sensitivity near the center of the distribution than at the tails. Unlike the one-sample KS test, it can be performed under more general conditions that allow for discontinuity, heterogeneity, and dependence across samples [31]. Of the two cases that showed the lowest tsKS statistic, it is interesting to note that case 4 demonstrated high agreement with measured intracranial pressure ($Diff_{ICPmd}$ 1.34–1.47 cm water). In contrast, Case 5 demonstrated a lower agreement ($Diff_{ICPmd}$ 5.8–6 cm water). The distribution of intracranial pressures in the training set likely impacted the model performance. A recognized limitation of decision tree algorithms, particularly since case 5 had a measured intracranial pressure (17 cm water) located at the tail of the left skewed intracranial pressure distribution in the XGB model training dataset [3, 32]. This was also demonstrated in the cautionary subset cases (8 and 9), where measured intracranial pressure was at a lower range of (11 and 8 cm water,

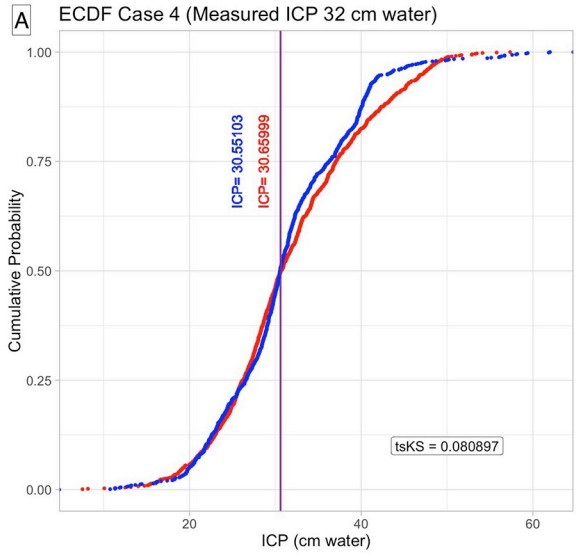

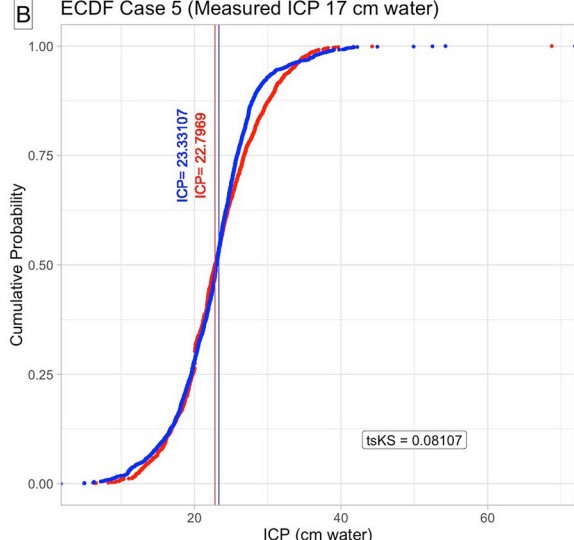

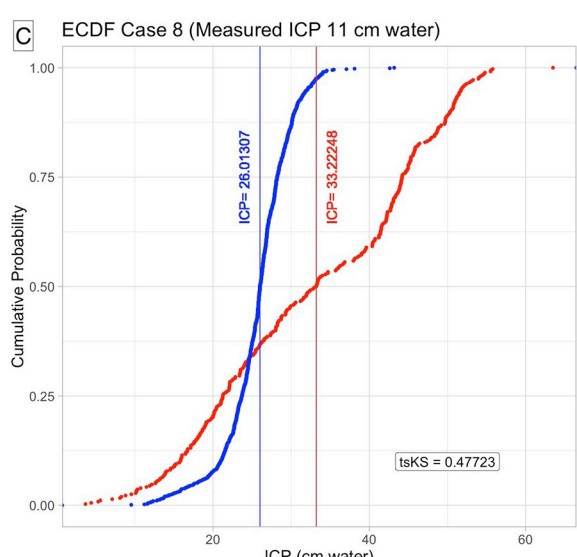

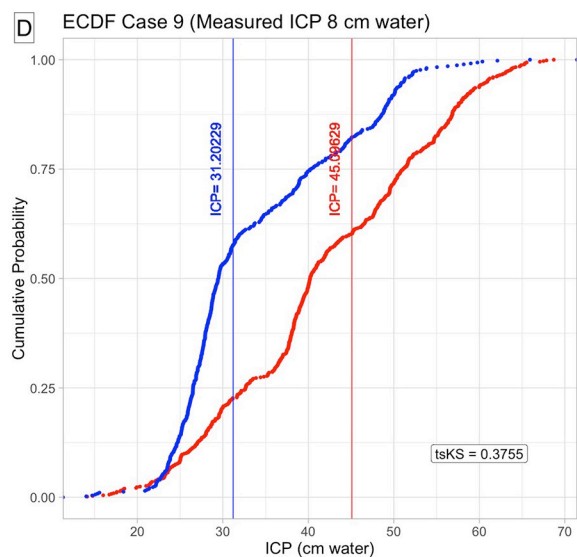

**Fig 5.** (A-D) A comparison of the empirical cumulative distribution function of intracranial pressure predictions derived from the Extreme Gradient Boost algorithm from four cases with contrasting two-sample Kolmogorov-Smirnov statistics from four cases. Two hold-out test cases demonstrating the lowest two-sample Kolmogorov-Smirnov statistic (tsKS). Cases 4 and 5 (A, B) demonstrate favorable concordance between venous and arterial derived predictions in contrast to the subset cases 8 and 9 (C, D), where the concordance is poor. The difference in separation of the ECDF between the two models can be observed in cases 8 and 9 (C, D). The tsKS statistic depends on a ratio parameter consisting of the product of the distribution data points divided by the sum [25]. Red = arterial model, Blue = venous model.

respectively). Moreover, the left-skewed distribution of measured intracranial pressure in the analyzed data set meant that the weight of the data points was in the body of the distribution ≥22 cm water (Fig 8). Hence, in the absence of sufficient training data, the model's predictive accuracy was degraded at lower intracranial pressure ranges <22 cm water.

The distribution shape parameters (skew, kurtosis, and $DI_{ICP}$) and $Diff_{ICPmd}$ had a strong negative correlation. Therefore, a more positive skew, kurtosis, and higher $DI_{ICP}$ were

**Table 3. Multivariate analysis of variance (MANOVA) estimating the associations between Diff$_{ICPmd}$ and both parameters of the imaging and distribution of Extreme Gradient Boost derived intracranial pressure predictions.**

|  | F value | p-value |
|---|---|---|
| Laterality | 8.5062 | 0.01009* |
| nIOP$_i$ | 5.1643 | 0.0372* |
| n | 2.3227 | 0.147 |
| Model | 0.1355 | 0.7176 |
| sd | 10.829 | 0.004608* |
| Kurtosis | 6.0616 | 0.02555* |
| Skew | 7.0544 | 0.01726* |
| DI$_{ICP}$ | 5.9705 | 0.02652* |
| ADS | 0.0787 | 0.7827 |
| KS | 2.9901 | 0.103 |
| tsKS | 5.1695 | 0.03712* |
| Distribution | 14.354 | 0.001611 |
| Case | 65.217 | <0.0001* |
| sEnt | 4.8744 | 0.0422* |

There was a strong association between whether imaging was conducted on both eyes (Laterality), the levels of induced intraocular pressure applied during the imaging (nIOP$_i$), distribution shape parameters (sd = standard deviation, skew, kurtosis, DI$_{ICP}$ = the definite integral ±1cm water of the median), concordance between venous and arterial distributions (tsKS = two sample Kolmogorov-Smirnov statistic), and Shannon entropy (sEnt) with the absolute difference between measured and predicted intracranial pressure (Diff$_{ICPmd}$). Statistical parameters that indicated convergence on a normal distribution (ADS = Anderson-Darling statistic, and KS = Kolmogorov-Smirnov statistic) and the number of data points tested (n) did not achieve statistical significance. Model = arterial/venous, Distribution = Type of distribution of the XGB derived prediction, Case = Case subtype hold-out / cautionary.

associated with a favorable agreement of predicted with measured intracranial pressure. The exception was with sd where a strong positive correlation with Diff$_{ICPmd}$ was consistent with an accurate prediction. Given that the probability density distribution is normalized, it should exhibit a unitary integral (area under the probability density distribution should = 1). Therefore, a higher DI$_{ICP}$ would equate to a lower dispersion of predictions around the median. Interestingly, DI$_{ICP}$ correlated more strongly with the absolute difference between measured and predicted intracranial pressure compared to tsKS. A possible explanation is that DI$_{ICP}$ depends on the convergence of one distribution and evaluates a narrow region of that distribution from which Diff$_{ICPmd}$ is computed. This was in contrast to tsKS, which is dependent on the total area of two model distributions and, theoretically, more sensitive to the entire distribution range, including the tail of the distribution, which is the outliers location.

The convergence to a uniform distribution was associated with a lower agreement with measured intracranial pressure, particularly for the arterial model. The venous predictions approximate the ideal log-normal distribution more closely than the arterial predictions. The current results suggest that a transition through log-normal and related skewed distributions such as the beta family, gamma, or logistic distributions may occur before convergence to a uniform distribution in the venous system. The dominance of skewed and, particularly, the log-normal distribution is consistent with the results of the central limit theorem, which states that a random variable that is the sum of many independent variables or variables with weak interactions converges on a Gaussian distribution. Likewise, a random variable resulting from a multiplicative product with synergistic and strong interactions of several variables converges on a log-normal distribution, which does not violate the central limit theorem as a log-normal

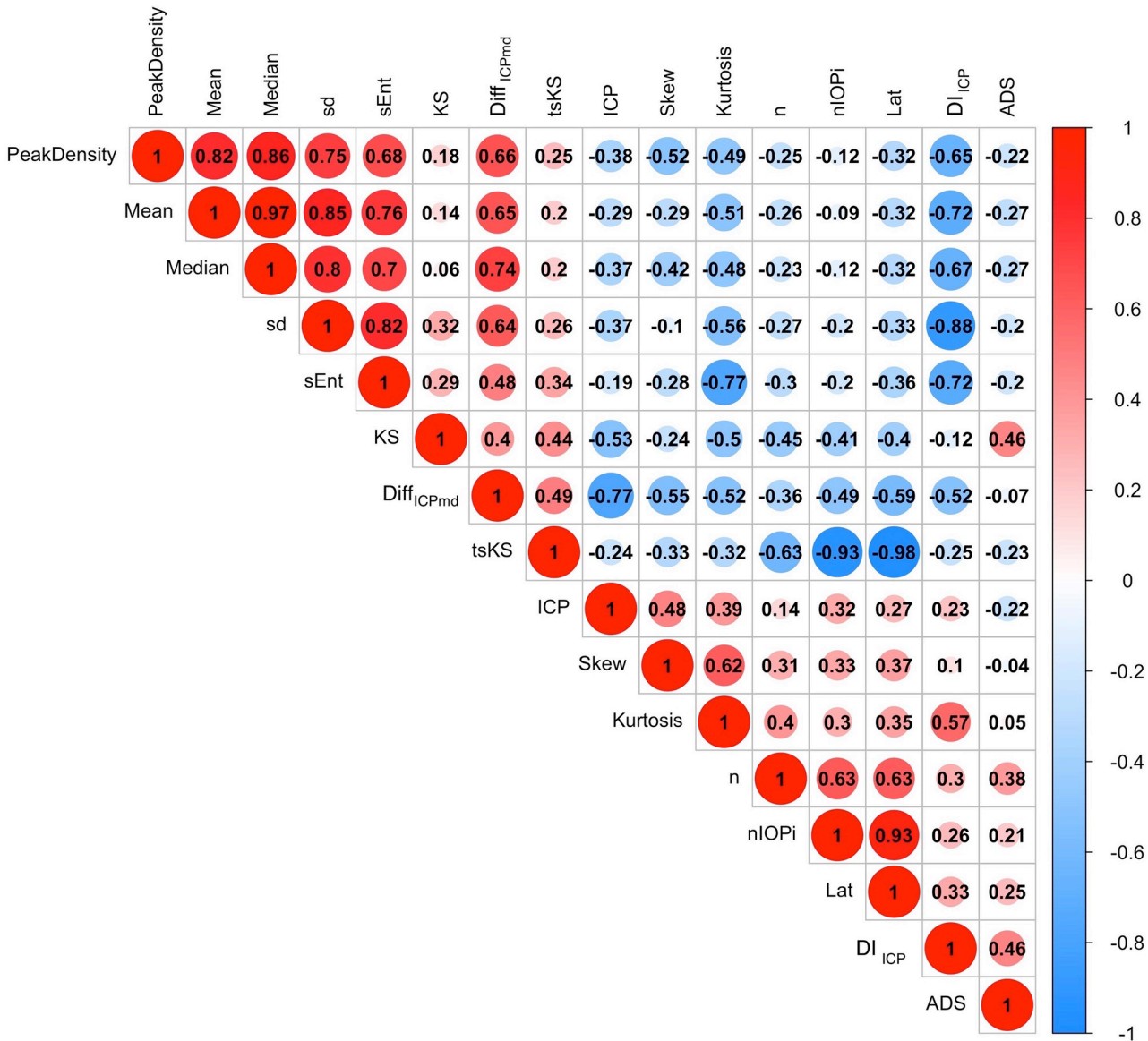

**Fig 6. Correlation matrix comparing distribution and imaging parameters from the hold-out and cautionary subsets.** Features from the top row are of interest. There was a significant negative correlation of $Diff_{ICPmd}$ with laterality (bilateral was numerically coded as 2 and unilateral was 1 for this analysis) of -0.59. There was a moderate to low correlation with parameters of the distribution of the XGB-derived prediction (ADS = Anderson-Darling statistic, KS = Kolmogorov-Smirnov statistic). However, the correlation with $DI_{ICP}$ = the definite integral ±1cm water of the median was strongly negative (-0.52), indicating that the higher the weight of the area under the curve within these bounds, the more accurate was the agreement between predicted and measured intracranial pressure. Similarly, the correlation with tsKS = two sample Kolmogorov-Smirnov statistic (0.49) was significant, indicating that the higher the overlap between the vascular model distributions, the higher the agreement with measured intracranial pressure. Comparably, Shannon entropy (sEnt) showed a strong positive correlation (0.48) indicating convergence to a uniform distribution (increased randomness) with higher $Diff_{ICPmd}$ values.$nIOP_i$ = the levels of induced intraocular pressure applied during the imaging, n = total number of tested data points.

distribution converges to a normal distribution in the logarithmic domain [8, 33, 34]. The majority of interactions in highly interconnected systems, especially in biological systems, are multiplicative and synergistic rather than additive. Therefore, a log-normal distribution results from this interaction [33]. This may explain the dominance of log-normal and skewed

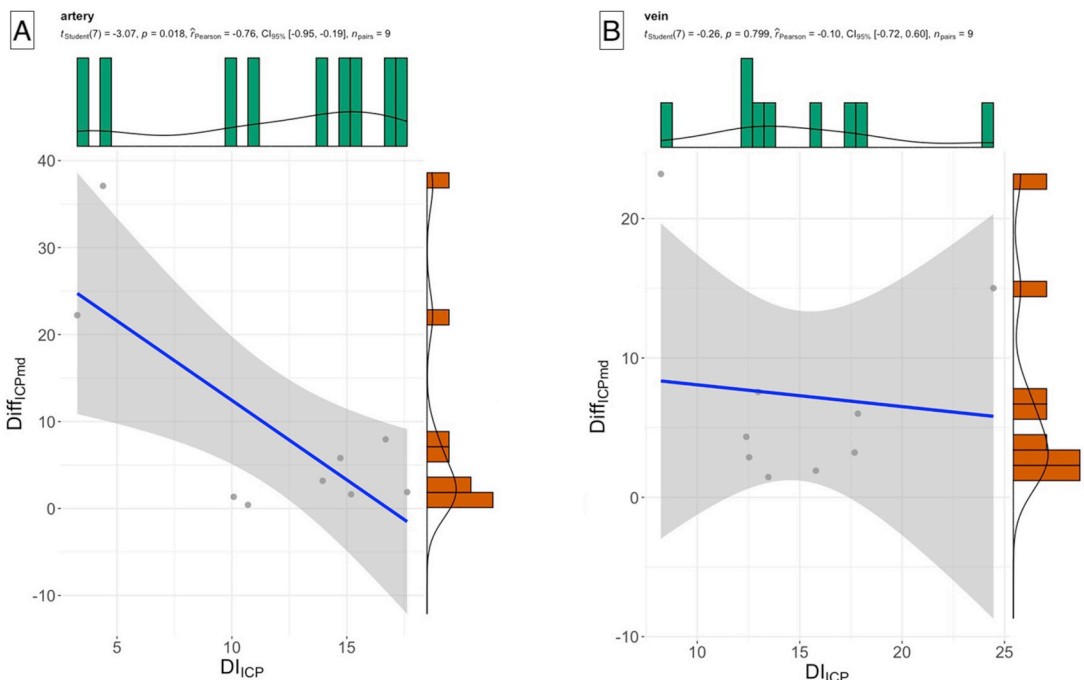

**Fig 7.** Pearson correlation between the definite integral ±1cm around the median of the probability density distribution (DI$_{ICP}$) and the absolute difference between predicted and measured intracranial pressure (Diff$_{ICPmd}$) for the A) arterial and B) venous models. Only A) the arterial model ($\hat{r}_{Pearson}$ = -0.76, p = 0.02) achieved statistical significance, in contrast to B) the venous model ($\hat{r}_{Pearson}$ = -0.10, p = 0.799). This indicated that the arterial model was a more discriminatory indicator of agreement between measured and predicted intracranial pressure.

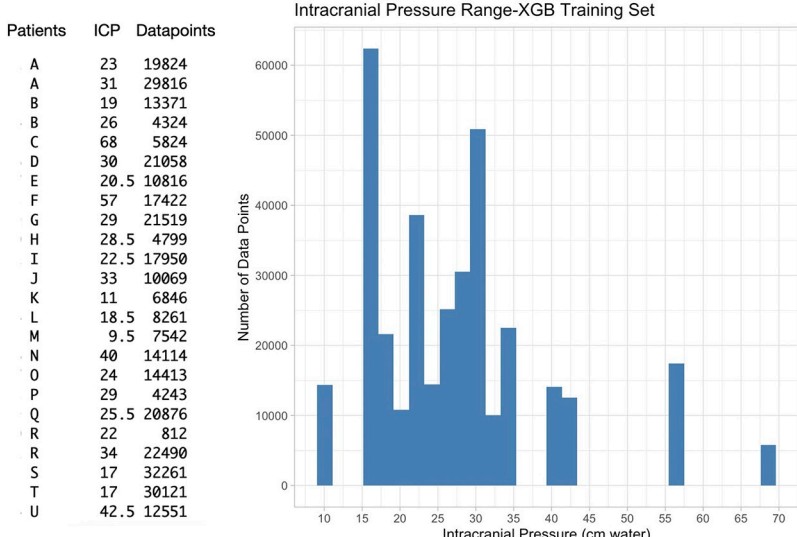

**Fig 8. Distribution of intracranial pressure and the number of analyzed image data points from the Extreme Gradient Boost training data set.** There is a low contribution of data points to the model at intracranial pressure levels <17 and >43 cm water with two participants below (K, M) and above (C, F) these boundaries, respectively. ICP = intracranial pressure in cm water [3].

distributions closely related to the log-normal distribution of intracranial pressures derived from the XGB algorithm. Strong non-linear dynamics dominated interactions between intracranial, intraocular pressure, and the retinal vascular pulse. In a recent publication, we used a linear mixed-effects model to correlate the harmonic amplitude distribution in the retinal vascular system with intraocular and intracranial pressure. This approach computed the variance of the model parameters, thereby quantitatively estimated their contribution to the pulsation dynamics at the optic disc in linear space. It was demonstrated that linear interactions of intraocular, intracranial pressure, and the retinal vascular pulse accounted for <9% of the variance. This was due to a non-constant variance (heteroscedasticity) of the retinal vascular pulse amplitude between individuals [35]. In previous work a generalized additive approach to define the geometry of the non-linear component of the interaction. This approach accounted for 49.21- 62.96% of the variance in the arterial and venous models, respectively [36]. Hence, non-linear synergistic dynamics between the retinal vascular pulse, intraocular, and intracranial pressure may account for the convergence on skewed distributions. This finding may also account for the outcome from the Bland-Altman analysis, where the highest agreement between measured and the median predicted intracranial pressure was observed in previous work. As the geometric mean of a log-normal distribution is equal to the median [3, 33], which therefore represents the ideal measure of central tendency for this type of skewed distribution.

Although the pulsation patterns and lumbar puncture measured intracranial in cases 8 and 9 were consistent with a normal intracranial pressure pattern, where venous pulsation amplitudes are higher compared to the arterial [35]. These cases highlight the challenges in non-invasive intracranial pressure estimation, where independent input variables can display a wide range of variance despite having similar intracranial pressures. Furthermore, the inconsistency in intracranial prediction in the latter two cases is also related to the inherent limitation of a decision tree algorithm. In contrast to a linear model, a recognized limitation of decision tree algorithms is its inability to extrapolate values by extension of the model boundaries. Therefore it may provide spurious results for values at the extremes of the training set values [32]. In both cases, 8 and 9 measured intracranial pressure were at the lower end of the normal range, and the Extreme Gradient Boost training set included only two subjects with a measured intracranial pressure over this range (Fig 8). Therefore, a more significant number of training examples would improve predictive accuracy for low intracranial pressure cases. However, a long observation period would be required based on the experience from the current study. The selective nature of modified photoplethysmography requires subject cooperation, which precludes imaging in a population of subjects with significant neurological disorders. Therefore, imaging was limited to participants with minimal neurological impairment. Moreover, lumbar puncture is invasive and unlikely to be performed based on a low clinical suspicion, further limiting case recruitment. Our research group is currently developing a handheld modified photoplethysmography solution to simplify data acquisition and address some of these restrictions.

The number of cases in this analysis was limited. A power analysis for medium effect size and statistical power of 80% showed that a population of 34 cases is required to confirm the $DI_{ICP}$ as a discriminatory parameter and minimize the Type II error (failing to reject a false null hypothesis). However, the initial analysis showed that it could supplement the PDF and potentially serve as an individualized statistical parameter of reliability of the XGB approach. The statistical power of machine learning algorithms demands a large sample size in the training set [37]. This allows the inclusion of case variants and extension of the range of interactions between the model variables. Several factors determine the volume of required training data, including model complexity, the complexity of the learning algorithm, label features of interest,

tolerable error margin, and variance in the input features. It remains unresolved how to best determine the sample size for a particular model when analyzing medical imaging data [38].

Differences in dynamic range in the pulsation parameters between the retinal veins and artery are highlighted both in Table 1 and Fig 2. These differences likely arise from structural and functional variability in the retinal vasculature and their characteristic interactions with pressurized anatomical chambers along their respective paths. The retinal arterioles lack an elastic lamina but possess a well-developed medial smooth muscle structure with 5–7 layers. In contrast, retinal veins have thinner walls (13.929±0.041 $\mu$m) compared to arteries (17.559 ±0.062 $\mu$m), and a thinner muscle layer of 3–4 layers that transitions to fibroblasts near the optic disc [39]. These structural differences influence the compliance and impedance characteristics of the retinal vasculature. While direct measurements of compliance and incremental modulus of elasticity ($E_{inc}$) in the retinal vasculature are lacking, inferences from systemic vessels can be drawn, where veins exhibit higher compliance, with a sigmoidal pressure-volume curve compared to the curvilinear relationships seen in arteries [40–45]. These differences in compliance likely lead to differences in pulsation characteristics. Although physiological interactions between the retinal vasculature with pressurized anatomical chambers: intracranial, intraorbital, and intraocular spaces are yet to be fully understood. Our recent work highlights previously unrecognized and clinically significant interactions of the retinal arterioles with intracranial pressure, evidenced by the ability to generate intracranial pressure predictions from the arterial tree with an accuracy comparable to that of the retinal veins [3]. However, phase relationships with intracranial pressure remain unknown in this part of the retinal vascular system. In contrast, the intracranial pressure wave may dominate the pulse frequencies in the structurally thinner and higher compliance retinal venous wall, potentially explaining the higher venous Fourier coefficients, higher $HRW_a$, and its attenuation characteristics [35, 46]. Observed differences in the phase of the retinal vascular pulse support this explanation, with experimental data showing phase congruence between retinal venous pulsation and intracranial pressure [47]. Future research in this field is recommended to shed light on the physiological interactions of the retinal vessels with anatomically related pressurized chambers.

In summary, the probability density distribution for the cautionary cases demonstrated unusual features. The arterial distribution in both cases converged on a uniform distribution, and the venous distribution of case 9 converged on a heavy-tailed beta distribution, all contributing to a low $DI_{ICP}$ statistic. The venous and arterial distributions in case 8 showed negative skews, contrary to all other cases in this study. The unusual shape parameters in case 8 resulted in the $DI_{ICP}$ statistic beyond two sd of the mean for both models. The probability density distribution shape parameters can therefore be significant indicators of the predictive accuracy of the machine learning model.

## Conclusions

Despite a limited-sized dataset, the results support a dual and complementary analysis approach from independently derived retinal arterial and venous intracranial pressure estimates. Additionally, the probability density distribution of the ICP predictions should ideally converge on a log-normal or related positively skewed distribution (beta, gamma, logistic) with a low sd, and high kurtosis indicating fewer outliers. Interestingly, Shannon entropy had the lowest correlation with $Diff_{ICPmd}$, among the tested distribution features. A larger area under the probability density curve ±1 cm water ($DI_{ICP}$) and the higher the concordance between the arterial and venous distributions as indicated by a lower tsKS statistic may provide individualized accuracy parameters of the machine learning predictive outcome and further support a clinical decision-making algorithm.

## Supporting information

**S1 Data.**
(CSV)

## Author Contributions

**Conceptualization:** Anmar Abdul-Rahman.

**Data curation:** Anmar Abdul-Rahman, William Morgan, Aleksandar Vukmirovic, Dao-Yi Yu.

**Formal analysis:** Anmar Abdul-Rahman.

**Investigation:** Anmar Abdul-Rahman.

**Methodology:** Anmar Abdul-Rahman.

**Project administration:** Anmar Abdul-Rahman, William Morgan, Aleksandar Vukmirovic, Dao-Yi Yu.

**Software:** Anmar Abdul-Rahman, William Morgan, Aleksandar Vukmirovic.

**Supervision:** William Morgan, Dao-Yi Yu.

**Validation:** William Morgan, Aleksandar Vukmirovic, Dao-Yi Yu.

**Visualization:** Anmar Abdul-Rahman, William Morgan, Aleksandar Vukmirovic, Dao-Yi Yu.

**Writing – original draft:** Anmar Abdul-Rahman.

**Writing – review & editing:** Anmar Abdul-Rahman, William Morgan, Aleksandar Vukmirovic, Dao-Yi Yu.

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
