## [Decision Letter · Decision Letter 0]

5 Feb 2024

PONE-D-23-31419Probability Density and Information Entropy of Machine Learning Derived Intracranial Pressure PredictionsPLOS ONE

Dear Dr. Abdul-Rahman,

Thank you for submitting your manuscript to PLOS ONE. After careful consideration, we feel that it has merit but does not fully meet PLOS ONE’s publication criteria as it currently stands. Therefore, we invite you to submit a revised version of the manuscript that addresses the points raised during the review process.

We look forward to receiving your revised manuscript.

Kind regards,

Alon Harris

Academic Editor

PLOS ONE

Journal Requirements:

   "I have read the journal's policy and the authors of this manuscript have the following competing interests: The authors Anmar Abdul-Rahman, William Morgan, Dao-Yi Yu are co-inventors of the Modified photoplethysmography device. We have no financial interest in the results of the study."

5. We note that your Data Availability Statement is currently as follows: All relevant data are within the manuscript and its Supporting Information files.

Reviewers' comments:

Reviewer's Responses to Questions

**Comments to the Author**

1. Is the manuscript technically sound, and do the data support the conclusions?

Reviewer #1: Yes

Reviewer #2: Partly

2. Has the statistical analysis been performed appropriately and rigorously? 

Reviewer #1: Yes

Reviewer #2: N/A

3. Have the authors made all data underlying the findings in their manuscript fully available?

Reviewer #1: Yes

Reviewer #2: Yes

4. Is the manuscript presented in an intelligible fashion and written in standard English?

Reviewer #1: Yes

Reviewer #2: No

5. Review Comments to the Author

Reviewer #1: This manuscript addresses the quantitative study of a dataset with the values of the intracranial pressure measurements using a statistical approach based on the concept of probability density function (PDF). The concordance between the arterial and venous probability density functions was estimated on a subset made of 7 hold-out test cases, containing retinal vascularpulse and intracranial pressure (ICP) measurements obtained using modified photoplethysmography and lumbar puncture, and for which good agreement was found between ICP data and ICP model predictions. Two cautionary subset cases (Case 8 and Case 9), for which disagreement was observed between measured and predicted ICP, were compared to the seven hold-out test cases. Despite the limited sample size, the Authors conclude that their results support the adoption of a statistical modeling tool for ICP predictions as a noninvasive complement to the invasive technique based on lumbar puncture.

The article addresses a central topic in the modern approach to mathematical and computational modeling of the eye and its pathologies, consisting of the use of quantitative statistical methods to mitigate the intrinsic uncertainty that affects the decision-making process based on a "black-box" strategy. The possibility to extract reliable information on a dataset by means of a simple visualization of a PDF and evaluation of its integral between two points in the probability space, looks very promising and appealing for a routine adoption in a clinical context.

This Reviewer, however, is concerned with some basic questions about the proposed methodology which should be successfully addressed by the Authors in a revised version of the article in order their manuscript to be considered for publication in the Journal "PlosOne".

1. Page 4, ten lines from the bottom.

"This is the case for intracranial pressure estimation, where a small margin of error would result in a significant difference in a clinical outcome."

This sentence seems rather important in the subsequent evaluation of model predictions, but there does not seem to be any motivation to support it. The Authors should expand this aspect of their presentation, possibly explaining if ICP is "more sensitive" to errors that other pressures in the eye (for instance, IOP), and, if so, why.

2. Page 8, bottom. Comment on Figure 2.

Results seem to clearly indicate that the violin plots of the veins display a wider dynamic range than those of arteries, in both hold-out and cautionary subsets. It would be very interesting to see whether it may be possible to connect this behavior to the fluid-mechanical (rather different) characteristics of arteries and veins. This specific issue belongs to the more general question of how to combine physiology and machine-learning techniques, to devise an optimal methodology in the treatment and analysis of human data.

3. Page 9, Table 1.

It is interesting to hear the Authors' opinion about whether the knowledge of more than two Fourier coefficients in the HWA may provide a better accuracy of the descriptive statistics. If so, as it appears to be reasonable, how difficult might be the extension of the present approach to the evaluation of the first N Fourier coefficients, N being the first integer such that the coefficient a_{N+1} (b_{N+1}) is smaller than a given tolerance.

MINOR ISSUES:

1. As a general comment, it appears that the acronym are rarely defined on their first occurrence, rather, their definition is made in the caption of the tables and/or figures. This approach seems to confer a touch of "confusion" to the presentation and, likely, could be amended by defining properly each quantity when it is mentioned for the very first time.

2. Page 3,

line 12 from top: "it is dependent on the" -- "is dependent on the"

4 lines from bottom: "weights should, by definition, should coincide." -- "weights should, by definition, coincide."

3. Page 4: "between a continuous random variable (X)": the letter "X" should be non-capitalized.

"The function (f(x)) is piecewise continuous.": why "(f(x))" and not "f(x)"??

4. Page 6: "A total of A total of 129,600 data points" -- "A total of 129,600 data points"

"conflicting predictions from the arterial and venous models and were found in ongoing evaluation of the model." -- "conflicting predictions from the arterial and venous models were found in ongoing evaluation of the model."

5. Page 10, line 1: "of the hold-out test cases. Whereas the median" -- "of the hold-out test cases, whereas the median"

6. Page 18, ten lines from bottom: "The analysis detailed in chapter ??" this unresolved reference must be fixed.

Reviewer #2: Recommendation is to include to a list of references and to study at least the following papers: S.T. van Hal et al . Neurocrit Care. https://doi.org/10.1007/s12028-023-01910-2 ; Jue Wang et al. BMC Medical Informatics and Decision Making. https://doi.org/10.1186/s12911-023-02247-8;

6. PLOS authors have the option to publish the peer review history of their article (what does this mean?). If published, this will include your full peer review and any attached files.

Reviewer #1: No

Reviewer #2: No

---

## [Author Response · Author response to Decision Letter 0]

9 Feb 2024

RESPONSE TO REVIEWERS PONE-D-23-31419: PROBABILITY DENSITY AND INFORMATION ENTROPY OF MACHINE LEARNING DERIVED INTRACRANIAL PRESSURE PREDICTIONS 

ANMAR ABDUL-RAHMAN, WILLIAM MORGAN, ALEKSANDAR VUKMIROVIC, DAO-YI YU 

We deeply appreciate the reviewers’ generous contribution of time, valuable insights, and expert feedback. Their thoughtful critique significantly improved the communication of our findings to clinical researchers in the field. We believe the results of this paper provide a statistical method to validate the output of a machine learning-generated intracranial pres- sure prediction on an individual case basis, with potential applications in the management of patients with neurological and ophthalmic disorders. We implemented the reviewers’ recommendations and hope that these changes meet the journal’s requirements. 

To facilitate the review, sentence line numbers are termed as Revised Line... to indicate the line number in the manuscipt document. 

We have also color coded the typesetting in our response letter: 

• The reviewer’s comment is in red. • The authors’ response is in black. • The changes in the manuscript in blue. 

The references and equations cited in this response are numbered in sequence as they appear in the current document. The numerical sequence has been adjusted accordingly in the manuscript. 

1 

Response to Reviewer #1: 

1. Page 4, ten lines from the bottom. ”This is the case for intracranial pressure estimation, where a small margin of error would result in a significant difference in a clinical outcome.” 

This sentence seems rather important in the subsequent evaluation of model predictions, but there does not seem to be any motivation to support it. The Authors should expand this aspect of their presentation, possibly explaining if ICP is ”more sensitive” to errors that other pressures in the eye (for instance, IOP), and, if so, why. 

We are grateful for this insightful comment from reviewer #1. We wish to take the oppor- tunity to expand on the recognised practical challenges in the precision and accuracy in ICP measurements. The following comment has been added to the introduction line 83 to 97: 

There is a wide variability in normal intracranial pressure values largely due to differences in age, gender, and body mass index. Although normal values of 7-15mmHg have been re- ported. [1] In a prospective study of lumbar puncture measured ICP in 339 normal subjects, Bø et al. reported a reference range of approximately 3-22mmHg. [2]. Their findings suggest that physiological ICP may vary by up to a 7-fold range. Additionally studies demonstrate variation of continuously measured ICP within individuals, likely due to the multitude of interactions between physiologic parameters involved in intracranial pressure homeostasis, including postural, cardiovascular, neurological and respiratory parameters. [3] In a sys- tematic review and meta-analysis of ICP monitoring systems Zaccharetti et al. found that the average error between simultaneous ICP measurements from different pressure sensors was approximately 1.5 mmHg, but the variability was large, with up to 11.4 mmHg differ- ence in 95% of readings. [4] These factors render determining the precision and accuracy of ICP measurements challenging even in absence of underlying pathology. 

2 

2. Page 8, bottom. Comment on Figure 2. 

Results seem to clearly indicate that the violin plots of the veins display a wider dynamic range than those of arteries, in both hold-out and cautionary subsets. It would be very in- teresting to see whether it may be possible to connect this behavior to the fluid-mechanical (rather different) characteristics of arteries and veins. This specific issue belongs to the more general question of how to combine physiology and machine-learning techniques, to devise an optimal methodology in the treatment and analysis of human data. 

We appreciate reviewer #1’s perceptive observation. We would like to discuss in further detail some reasons behind the differences in the dynamic range of pulsation values in the retinal vasculature. The following comment has been added to the discussion lines 511 to 539: 

Differences in dynamic range in the pulsation parameters between the retinal veins and artery are highlighted both in Table 1 and Fig 2. These differences likely arise from structural and functional variability in the retinal vasculature and their characteristic in- teractions with pressurized anatomical chambers along their respective paths. The retinal arterioles lack an elastic lamina but possess a well-developed medial smooth muscle struc- ture with 5-7 layers. In contrast, retinal veins have thinner walls (13.929±0.041 μm) compared to arteries (17.559 ±0.062 μm), and a thinner muscle layer of 3-4 layers that transitions to fibroblasts near the optic disc [5]. These structural differences influence the compliance and impedance characteristics of the retinal vasculature. While direct measure- ments of compliance and incremental modulus of elasticity (Einc) in the retinal vasculature are lacking, inferences from systemic vessels can be drawn, where veins exhibit higher com- pliance, with a sigmoidal pressure-volume curve compared to the curvilinear relationships seen in arteries [6–11]. These differences in compliance likely lead to differences in pulsation 

characteristics. Although physiological interactions between the retinal vasculature with 

3 

pressurized anatomical chambers: intracranial, intraorbital, and intraocular spaces are yet to be fully understood. Our recent work highlights previously unrecognized and clinically significant interactions of the retinal arterioles with intracranial pressure, evidenced by the ability to generate intracranial pressure predictions from the arterial tree with an accuracy comparable to that of the retinal veins [12]. However, phase relationships with intracra- nial pressure remain unknown in this part of the retinal vascular system. In contrast, the intracranial pressure wave may dominate the pulse frequencies in the structurally thinner and higher compliance retinal venous wall, potentially explaining the higher venous Fourier coefficients, higher HRWa, and its attenuation characteristics [13,14]. Observed differences in the phase of the retinal vascular pulse support this explanation, with experimental data showing phase congruence between retinal venous pulsation and intracranial pressure. [15] Future research in this field is recommended to shed light on the physiological interactions of the retinal vessels with anatomically related pressurized chambers. 

3. Page 9, Table 1. 

It is interesting to hear the Authors’ opinion about whether the knowledge of more than two Fourier coefficients in the HWA may provide a better accuracy of the descriptive statistics. If so, as it appears to be reasonable, how difficult might be the extension of the present approach to the evaluation of the first N Fourier coefficients, N being the first integer such that the coefficient aN+1 bN+1 is smaller than a given tolerance. 

Reviewer #1 raises a fascinating question regarding the limit of harmonics that can be detected from the retinal circulation. Although theoretically there are an infinite number of Fourier harmonics that can be extracted from a signal, practically this is limited by both the finite signal duration and the signal-to-noise ratio. For a practical signal, only a finite window in time can be considered, limiting the highest frequency that can be accurately 

4 

represented using Fourier analysis. This relationship is encapsulated by the Heisenberg- Gabor uncertainty principle [16]. The principle asserts that there is a tradeoff between temporal (∆t) and spectral (∆f) resolution, which is represented by the mathematical identity ∆f · ∆t ≥ C, where C is a constant. [17] Accordingly, the shorter the time win- dow, the higher the detectable frequencies, but this comes at a cost to accuracy. Moreover, as the signal-to-noise ratio decreases (more noise), the higher frequency harmonics become increasingly unreliable due to noise contamination. At some point, the higher frequencies become indistinguishable from pure noise and convey no meaningful information about the original signal. Therefore, in practice, the effective limit of the number of harmonics that can be considered is based on signal duration and the signal-to-noise ratio to ensure reliable results. 

By extension from what is known about the harmonics of the systemic circulation, which have been studied more thoroughly compared to the ophthalmic circulation, it has been re- ported that the first two harmonics account for approximately 85% of the pulsatile portions of pressure and flow waves in large arteries [18]. Krovetz et al. analyzed the harmonics of intracardiac and arterial pressure waves, reporting that 3-10 harmonics were required to re- produce measured pressure waveforms. However, theoretically derived differential pressures and derivatives appeared to contain six-fold the number of higher-frequency harmonics [19]. Despite this theoretical estimate, there is a fundamental limit to the number of harmonics extracted from measured pressure-flow waves. Beyond the tenth harmonic component, the harmonic magnitudes of the aortic pressure pulse and aortic flow pulse are negligible due to sampling factors and signal noise. A further limit is imposed by the Nyquist criterion, which requires a sampling frequency of at least twice the highest frequency content to accurately reconstruct the original waveform. Due to a low signal-to-noise ratio, Milnor demonstrated that harmonics above 25 Hz are unreliable. Practically, the noise level and 

5 

resolution of recording instruments reduce the accuracy of harmonics beyond the sixth, and measurements are, therefore, subject to uncertainty at higher harmonics [20–23]. There are other considerations specific to the retinal circulation; the vessel wall and fluid viscosities attenuate the pulse wave with centrifugal propagation. The attenuation, or damping, is more significant at higher frequencies, a phenomenon known as dispersion. For example, the incisura is progressively damped as it propagates. Indeed, when the pulse reaches the femoral artery, the characteristic high-frequency features of the aortic pulse disappear, and instead, a smooth waveform is seen when the pulse reaches the arterioles. Damping results in an approximately sinusoidal waveform; therefore, higher harmonics may become increasingly negligible peripherally. Specific to ophthalmic imaging, optical and motion ar- tifacts coupled with the low amplitude vascular pulse impose further limits on the Fourier harmonics that can be effectively extracted from the retinal circulation [24]. 

MINOR ISSUES:. 1. As a general comment, it appears that the acronym are rarely defined on their first occurrence, rather, their definition is made in the caption of the tables and/or figures. This approach seems to confer a touch of ”confusion” to the presentation and, likely, could be amended by defining properly each quantity when it is mentioned for the very first time. 

We apologize for this oversight. The abbreviations have been reviewed and clarifications have been applied accordingly particularly in the materials and methods section. 

2. Page 3, line 12 from top: ”it is dependent on the” —-> ”is dependent on the” 4 lines from bottom: ”weights should, by definition, should coincide.” —-> ”weights should, by definition, coincide.” Corrected line 42. Corrected line 57. 

6 

3. Page 4: ”between a continuous random variable (X)”: the letter ”X” should be non- capitalized. ”The function (f(x)) is piecewise continuous.”: why ”(f(x))” and not ”f(x)”?? Corrected line 71. 

Corrected line 72 and 75. 

4. Page 6: ”A total of A total of 129,600 data points” —-> ”A total of 129,600 data points” ”conflicting predictions from the arterial and venous models and were found in ongoing evaluation of the model.” —-> ”conflicting predictions from the arterial and venous mod- els were found in ongoing evaluation of the model.” 

Corrected line 161. Corrected line 169. 

5. Page 10, line 1: ”of the hold-out test cases. Whereas the median” —-> ”of the hold-out test cases, whereas the median” The punctuation has been corrected line 256. 

6. Page 18, ten lines from bottom: ”The analysis detailed in chapter ??” this unresolved reference must be fixed. The reference has been corrected line 470. 

7 

Response to Reviewer #2: 

Reviewer #2: Recommendation is to include to a list of references and to study at least the following papers: S.T. van Hal et al . Neurocrit Care. https://doi.org/10.1007/s12028- 023-01910-2 ; Jue Wang et al. BMC Medical Informatics and Decision Making. https://doi.org/10.1186/s12911-023-02247-8;

We would like to thank reviewer #2 for drawing our attention to recent publications in the field we have added the following comment in the introduction lines 97 to 114: 

While evidence suggests artificial intelligence (AI) algorithms could aid clinical decision- making, current research emphasizes the need for systems enabling model evaluation, bias detection, and generalizability. Wang et al. compared machine learning models to tradi- tional scoring tools for predicting mortality risk in traumatic brain injury patients. Among 47 studies with 156 models, machine learning models showed relatively high accuracy for both in-hospital and out-of-hospital mortality. Notably, traditional tools achieved compa- rable accuracy. The authors highlight the need for standardized reporting and validation of machine learning models to ensure clinical applicability and generalizability. [25] Simi- larly, van Hal et al. assessed bias risk and clinical readiness of studies using AI to predict intracranial hypertension in traumatic brain injury patients. They found most studies had high bias risk and low readiness for clinical integration. Despite promising potential, the authors concluded further improvement and validation are necessary before implementing these models in clinical practice. [26] These findings underline the practical importance of bias detection, as AI algorithms may perform well under specific conditions not replicable in clinical settings. Using properties of the probability density function the approximation of the estimate to the ground truth may be confirmed by the capacity to validate ICP readings, irrespective of the measurement methodology. 

8 

1. Other Corrections 

 . (1)  Line 147 mentions written consent was obtained from participants.  

 . (2)  Line 157 ethics committee project number added.  

 . (3)  Line 200 the variable x in the equation changed to lowercase.  

 . (4)  Line 376 ”Whereas in the venous model...” changed to ”While in the venous  

model...” 

9 

References 

 1. Munakomi S, Das M. Intracranial Pressure Monitoring. StatPearls [Internet]. 2019;.  

 2. Bø SH, Lundqvist C. Cerebrospinal fluid opening pressure in clinical practice–a prospective study.  Journal of Neurology. 2020;267:3696–3701.  

 3. Jonas JB, Wang N, Yang D, Ritch R, Panda-Jonas S. Facts and myths of cerebrospinal fluid pressure  for the physiology of the eye. Prog Retin Eye Res. 2015;46:67–83.  

 4. Zacchetti L, Magnoni S, Di Corte F, Zanier ER, Stocchetti N. Accuracy of intracranial pressure  monitoring: systematic review and meta-analysis. Crit care. 2015;19:1–8.  

 5. Hogan MJ, Feeney L. The ultrastructure of the retinal blood vessels: I. The large vessels. J Ultrastruct  Res. 1963;9(1-2):10–28. doi:https://doi.org/10.1016/s0022-5320(63)80033-7.  

 6. Nichols WW, Edwards DG. Arterial elastance and wave reflection augmentation of systolic blood pressure: deleterious effects and implications for therapy. J Cardiovasc Pharmacol Ther. 2001;6(1):5–  21. doi:https://doi.org/10.1177/107424840100600102.  

 7. Moreno AH, Katz AI, Gold LD, Reddy R. Mechanics of distension of dog veins  

and other very thin-walled tubular structures. Circ Res. 1970;27(6):1069–1080. 

doi:https://doi.org/10.1161/01.RES.27.6.1069. 

 8. Keener J, Sneyd J. The circulatory system. In: Mathematical Physiology II: Systems Physiology.  New York: Springer-Verlag; 2009. p. 471–522.  

 9. Feher JJ. Vascular function: Hemodynamics. In: Feher JJ, editor. Quantitative Human Physi

---

## [Decision Letter · Decision Letter 1]

11 Jun 2024

Probability Density and Information Entropy of Machine Learning Derived Intracranial Pressure Predictions

PONE-D-23-31419R1

Dear Dr. Anmar Abdul-Rahman,

We’re pleased to inform you that your manuscript has been judged scientifically suitable for publication and will be formally accepted for publication once it meets all outstanding technical requirements.

Kind regards,

Alon Harris

Academic Editor

PLOS ONE

Additional Editor Comments (optional):

Reviewers' comments:

Reviewer's Responses to Questions

**Comments to the Author**

1. If the authors have adequately addressed your comments raised in a previous round of review and you feel that this manuscript is now acceptable for publication, you may indicate that here to bypass the “Comments to the Author” section, enter your conflict of interest statement in the “Confidential to Editor” section, and submit your "Accept" recommendation.

Reviewer #1: All comments have been addressed

Reviewer #3: All comments have been addressed

2. Is the manuscript technically sound, and do the data support the conclusions?

Reviewer #1: Yes

Reviewer #3: Yes

3. Has the statistical analysis been performed appropriately and rigorously? 

Reviewer #1: Yes

Reviewer #3: Yes

4. Have the authors made all data underlying the findings in their manuscript fully available?

Reviewer #1: Yes

Reviewer #3: Yes

5. Is the manuscript presented in an intelligible fashion and written in standard English?

Reviewer #1: Yes

Reviewer #3: Yes

6. Review Comments to the Author

Reviewer #1: (No Response)

Reviewer #3: All the comments and remarks of the previous reviewers were looked at and corrected and required discussion, information and literature was added.

7. PLOS authors have the option to publish the peer review history of their article (what does this mean?). If published, this will include your full peer review and any attached files.

Reviewer #1: **Yes: **Riccardo Sacco

Reviewer #3: No

---

## [Editor Report · Acceptance letter]

21 Jun 2024

PONE-D-23-31419R1 

PLOS ONE

Dear Dr. Abdul-Rahman, 

I'm pleased to inform you that your manuscript has been deemed suitable for publication in PLOS ONE. Congratulations! Your manuscript is now being handed over to our production team.

Kind regards, 

on behalf of

Dr. Alon Harris 

Academic Editor

PLOS ONE